# Infection Prevention Performance among In-Flight Cabin Crew in South Korea

**DOI:** 10.3390/ijerph18126468

**Published:** 2021-06-15

**Authors:** Jaegeum Ryu, Jungha Kim, Smi Choi-Kwon

**Affiliations:** 1College of Nursing, Seoul National University, Seoul 03080, Korea; ryu301@snu.ac.kr (J.R.); smi@snu.ac.kr (S.C.-K.); 2The Research Institute of Nursing Science, Seoul National University, Seoul 03080, Korea; 3Aeromedical Center, KoreanAir, Seoul 07505, Korea

**Keywords:** COVID-19, pandemic, infection control, inflight transmission, aircraft

## Abstract

COVID-19 was declared a worldwide pandemic in 2020; thus, preventing in-flight infection transmission is important for stopping global spread via air travel. Infection prevention (IP) performance among aircraft cabin crew is crucial for preventing in-flight transmission. We aimed to identify the level of IP performance and factors affecting IP performance among aircraft cabin crew during the COVID-19 pandemic in South Korea. An online survey was conducted with 177 cabin crew members between August and September 2020. The survey assessed IP performance, and IP awareness, using a five-point Likert scale, and also evaluated simulation-based personal protective equipment (PPE) training experience, and organizational culture. The average IP performance score was 4.56 ± 0.44. Although the performance level for mask-wearing was high (4.73 ± 0.35), hand hygiene (HH) performance (4.47 ± 0.56) was low. Multivariate analysis showed that IP performance was significantly associated with IP awareness (*p* < 0.05) and simulation-based PPE training experience (*p* < 0.05). Since HH performance was relatively low, cabin crew and airlines should make efforts to improve HH performance. Furthermore, a high level of IP awareness and PPE training experience can improve IP performance among cabin crew members. Therefore, simulation-based PPE training and strategies to improve IP awareness are essential for preventing in-flight infection transmission.

## 1. Introduction

Coronavirus disease 2019 (COVID-19) was first identified as pneumonia of unknown cause in China in December 2019 [1], and has since spread rapidly worldwide [2]. South Korea reported its first COVID-19 infection at the beginning of the global spread, due to the country’s close geographical proximity to China [3].

COVID-19 was largely spread from China through people traveling via aircraft, leading to community transmission in other countries [4]. Strong measures, such as border closures, air traffic restrictions, and citywide lockdowns have been implemented to stop the spread of COVID-19 within and between countries [5]. As borders closed, it became difficult for citizens in foreign countries to return home, to which some countries responded by operating national chartered flights to bring back their own citizens [6,7].

However, air travel was reported to have accelerated the H1N1 novel influenza pandemic in 2009. Therefore, in-flight infection prevention (IP) was crucial to stop the spread of the COVID-19 pandemic [8]. Effective strategies suggested to prevent infectious disease associated with air travel include entry screening, isolation, quarantine, and personal hygiene [9]. During a flight, viral transmission could occur by inhaling suspended infectious droplets or through direct contact with an unidentified person with COVID-19 who was asymptomatic while onboard the aircraft [10,11]. Moreover, people moving through the corridor of a confined cabin space, such as when cabin crew members provide in-flight service, could increase the infection risk [12,13]. To prevent the in-flight transmission of COVID-19, cabin crew members are advised to perform IP behaviors, such as using hand sanitizer for thorough hand hygiene (HH) and wearing appropriate personal protective equipment (PPE), such as N95 facemasks [14,15]. N95 facemasks have been reported to significantly reduce the risk of droplet transmission among cabin crew members and passengers onboard aircraft [16], and mass COVID-19 events on flights were rapidly reduced when mandatory mask-wearing was implemented on aircraft due to the significant transmission from pre-symptomatic and symptomatic individuals [17].

Previous studies have reported that IP awareness is related to IP performance [18,19]. For example, improved IP awareness has been shown to increase the intention to comply with IP guidelines among healthcare workers (HCWs) [20]. In particular, in 2015, South Korea experienced an outbreak of another type of coronavirus disease, Middle East Respiratory Syndrome (MERS), which necessitated that HCWs increase their IP awareness [21,22]. After the MERS outbreak, people came to recognize the importance of infection control measures such as hand hygiene and mask-wearing to prevent the spread of infectious disease in South Korea [22]. Along with IP awareness, previous research has indicated that simulation-based PPE putting on and removing may be related to improved IP performance [23]. Realistic and actual practice in properly putting on and removing PPE can ensure that HCWs improve their IP awareness in order to prepare for future uncertain situations during a pandemic [23,24].

Another factor reported to influence IP performance is organizational culture [25]. Organizational culture has been found to improve IP performance in ensuring safety against healthcare-associated infections [26]. In South Korea, organizational culture tends to be more authoritative and collectivist than in Western cultures [27,28]. Additionally, among cabin crew, a collectivist culture has been reported to be fostered due to job characteristics related to ensuring passengers’ in-flight service satisfaction and safety in a limited cabin space [29].

Thus, it is essential for cabin crew to engage in IP performance to prevent global transmission of infectious diseases via air travel. Further, it is necessary for cabin crew to strictly adhere to IP guidelines, due to possible exposure to various infectious agents during a flight. However, to the best of our knowledge, IP performance among cabin crew has not been investigated. Moreover, the factors that affect IP performance among cabin crew have not yet been clarified. Therefore, we aimed to increase understanding of the status of IP performance by aircraft cabin crew during the COVID-19 pandemic to identify the factors that affect IP performance.

## 2. Materials and Methods

### 2.1. Study Design

This was a cross-sectional, descriptive, investigative study conducted to better understand the level and related factors of IP performance among cabin crew during the COVID-19 pandemic. This study was conducted after receiving approval from the Institutional Review Board of Seoul National University (SNU 20-07-039).

### 2.2. Participants and Data Collection

This study was conducted with 184 cabin crew members of a major South Korean airline. Participants without in-flight service experience during the COVID-19 pandemic were excluded. Data were collected between August and September 2020, using an online survey (http://en.surveymonkey.com accessed on 29 August 2020). All participants were informed of the study’s purpose, content, and data collection procedures, and assured that their information would remain confidential and the collected data would not be used for purposes other than the study. Participants could only complete the online survey after providing their consent to participate in the study. Although the possibility of infection may differ depending on factors such as the type of aircraft, number of passengers, and flight time, the airline included in this study was operating the standard IP procedures at the time of data collection. The air inside each aircraft of this airline is circulated every 2–3 min through HEPA filters, and the cabin crew members are asked to wear masks, goggles, gloves, and gowns and to apply enhanced IP procedures during each flight. [30].

### 2.3. Measurement Variables

#### 2.3.1. General Participant Characteristics

Data were collected on the participants’ general characteristics, including gender, age, marital or partnership status, educational background, job position, and employment duration. We investigated whether the participants had previous experience handling passengers with confirmed or suspected cases of MERS or COVID-19 and quarantining after exposure to these diseases. Regarding COVID-19, according to Korean CDC guidelines, having quarantine experience also meant taking a test for COVID-19 if the participants had come in contact with infected persons or showed symptoms characteristic of COVID-19.

#### 2.3.2. IP Performance

To measure IP performance among cabin crew, we modified a tool developed by Hong et al. [31], which was originally developed to measure performance of standard precautions in nursing college students. The questionnaire comprised three areas with a total of 22 questions: HH (10 questions), mask-wearing (6 questions), and management of passengers suspected of having COVID-19 (6 questions). Participants responded using a five-point Likert scale (never performed = 1, rarely performed = 2, sometimes performed = 3, often performed = 4, always performed = 5). A higher value indicates higher performance. If participants had no experience with a particular item, they were asked to respond with “never experienced.”

We measured the tool’s content validity index (CVI) based on three infection control nurses, two airline nurses, and one cabin crew member, all of whom recorded an item-level CVI (I-CVI) of 0.78 or higher and a scale-level CVI/average (S-CVI/Ave) of 0.92, which are considered suitable [32]. The original tool [31] had a Cronbach’s α value of 0.95, whereas the tool used in our study had a value of 0.89.

#### 2.3.3. IP Awareness

To measure IP awareness among cabin crew, we modified a measurement tool developed by Hong et al. [31] comprising 22 questions covering the same three areas as the IP performance measurement tool, with items distributed in the same manner. Participants responded using a five-point Likert scale (not at all important = 1, not important = 2, neutral = 3, important = 4, and very important = 5); the higher the score, the higher is the awareness.

As with the IP performance measurement, the CVI of this questionnaire was assessed by three infection control nurses, two airline nurses, and one cabin crew, all of whom recorded an I-CVI of 0.78 or higher and an S-CVI/Ave of 0.97, which are considered suitable [32]. The original tool [31] had a Cronbach’s α value of 0.97, whereas the tool used in our study had a value of 0.82.

#### 2.3.4. Organizational Culture

To measure organizational culture, we modified a tool originally developed by Kim [33] to measure organizational culture in hospitals. It consists of seven questions, to which the participants respond using a seven-point Likert scale (strongly disagree = 1, often disagree = 2, slightly disagree = 3, neutral = 4, slightly agree = 5, mostly agree = 6, strongly agree = 7); the higher the value, the better is the organizational culture.

The CVI was measured based on three infection control nurses, two airline nurses, and one cabin crew member, all of whom recorded an I-CVI of 0.78 or higher and an S-CVI/Ave of 0.91, which are considered suitable [32]. The original tool [33] had a Cronbach’s α value of 0.85, whereas the tool used in our study had a value of 0.80.

#### 2.3.5. Simulation-Based PPE Training

Since PPE training experience has been shown to be related to improved IP performance, we also evaluated whether participants had previously received simulation-based PPE training. The airline provided guidelines for putting on and taking off PPE to the cabin crew online; however, to ensure IP performance, cabin crew members received simulation-based PPE training by an epidemiological intelligence service officer before boarding COVID-19-related chartered flights to Wuhan, China, and Milan, Italy in January and March 2020, respectively.

### 2.4. Statistical Analysis

Data were analyzed using SPSS (version 25.0; IBM Corp.). Participants’ general characteristics were analyzed using descriptive statistics (frequency, percentage, mean, and standard deviation), whereas their IP awareness and performance were analyzed using means and standard deviations. The difference between IP awareness and performance was analyzed using a paired *t*-test. IP performance and general characteristics were analyzed using independent *t-*tests or analysis of variance (ANOVA). The relationships between simulation-based PPE training experience and actual experience handling passengers with, or quarantining due to exposure to, a suspected or confirmed case of MERS and/or COVID-19 were analyzed using the Mann−Whitney U test (*n* < 30 cases). Correlations between IP performance, IP awareness, and organizational culture were analyzed using Pearson’s correlation coefficients. The factors affecting IP performance were analyzed using linear regression analysis with the bootstrap method, which is useful for data that do not meet the assumption of error normality [34]. For all measurement tools, reliability was analyzed using Cronbach’s α. The level of statistical significance was set at *p* < 0.05.

## 3. Results

### 3.1. Participant Characteristics

Of the 184 cabin crew members who agreed to participate in the study, 177 (96%) were included in the analysis. The participants included 133 women (75.1%), 131 college graduates (74.0%), and 105 general flight attendants (59.3%). Their average age was 37.58 ± 8.46 years, the largest proportion comprising people in their 40s (*n* = 69; 39.0%). The average working experience was 6.26 ± 3.38 years. There were 25 (14.1%) and 14 participants (7.9%) with experience of handling passengers with suspected and confirmed cases of COVID-19 and MERS in 2015, respectively. There were 19 (10.7%) and 3 (1.7%) participants with quarantine experience due to coming in contact with passengers with suspected and confirmed cases of COVID-19 or MERS (in 2015), respectively. There were 23 participants (13.0%) who had simulation-based PPE training experience (Table 1).

### 3.2. IP Performance and Awareness

The average level of IP performance was 4.56 ± 0.44. The subcategory with the highest score was handling passengers with suspected or confirmed cases of COVID-19 (4.90 ± 0.41), followed by mask-wearing (4.73 ± 0.35) and HH (4.47 ± 0.56). The average level of IP awareness was 4.75 ± 0.28. The subcategory with the highest score was handling passengers with suspected or confirmed cases of COVID-19 (4.97 ± 0.08), followed by mask-wearing (4.78 ± 0.35) and HH (4.61 ± 0.45).

Upon comparison, the level of IP performance (4.56 ± 0.44) was significantly lower (*p* < 0.05) than that of IP awareness (4.75 ± 0.28). By subcategory, the level of performance for HH (4.47 ± 0.56) was significantly lower than that of awareness (4.61 ± 0.45; *p* < 0.05) (Table 2).

### 3.3. Factors Related to IP Performance

A univariate analysis showed that simulation-based PPE training experience (*p* < 0.05), IP awareness (*p* < 0.05), and organizational culture (*p* < 0.05) affected IP performance. The performance level of cabin crew members with simulation-based PPE training experience (*n* = 23, median = 4.76, range: 3.33~5.00) was significantly higher than that of cabin crew members without such experience (*n* = 154, median = 4.64, range: 2.79~5.00). The performance of cabin crew members who experienced MERS in 2015 (*n* = 14, median = 4.87, range: 4.19~5.00) was higher than that of those who had no such experience (*n* = 163, median = 4.67, range: 2.79~5.00); however, the difference was not statistically significant (*p* = 0.078; Table 3).

The multivariate analysis showed that IP awareness (B = 0.77, *p* < 0.05) and simulation-based PPE training experience (B = 0.25, *p* < 0.05) significantly affected cabin crew members’ IP performance. The regression model had an explanatory power of 29%, which corresponded to the medium effect size as given by Cohen [35,36], and an appropriate Durbin−Watson d value of 1.97. All variance inflation factors were less than 10, thereby ruling out multi-collinearity [37]. We used the bootstrap method to perform a linear regression analysis, as the assumptions of multivariate normality, linearity, and normally distributed errors were not met (Kolmogorov−Smirnov normality test Z = 1.95, *p* < 0.05; Breusch Pegan test χ^2^ = 16.80, *p* < 0.05; Table 4) [34].

## 4. Discussion

This study investigated IP performance and its related factors among cabin crew members who had come into direct contact during a flight with passengers who were confirmed or suspected cases of COVID-19. It was found that cabin crew members’ IP performance during the COVID-19 pandemic was high, especially for mask-wearing. Additionally, IP awareness and simulation-based PPE training experience were found to be associated with IP performance.

The average IP performance score of cabin crew was 4.56, which was higher than that of medical staff (4.31) [33]. This may have been due to the timing of data collection. Unlike in previous studies, data in the present study were collected during the COVID-19 pandemic. It is likely that cabin crew members could have perceived the infection risk related to possible contact with passengers with confirmed COVID-19 cases during a flight as being more serious, thereby leading to increased IP performance. In fact, we found that cabin crew members who had experienced MERS in 2015 had better IP performance than those who had not (4.87 vs. 4.67).

However, the IP performance of cabin crew members in this study was also higher than that of nurses during the COVID-19 pandemic (4.28) as reported in another study [21]. This may be related to the unique Korean organizational culture among cabin crew, as Korea has a strong collectivist organizational culture [27,28]. Furthermore, cabin crew members work as a team during flights, and all procedures related to in-flight services, including attire, need to follow company guidelines. In particular, this airline emphasizes that the team leader must check the health condition of the cabin crew, such as whether they have a fever or respiratory symptoms, and whether they are wearing a mask before and during flights. In a further sub-analysis, we found that supervisors’ encouragement to comply with IP guidelines was highly correlated with IP performance among cabin crew (rho = 0.21, *p* < 0.05). As cabin crew members are influenced by their supervisor’s leadership when working [38], the role of team leaders is considered to be important to ensure that cabin crew comply with IP guidelines.

We found that, among the factors related to IP performance, the average score for mask-wearing was 4.73, which was consistent with previous studies conducted with emergency room medical staff during the SARS epidemic and among the general population during the COVID-19 pandemic [39,40]. This may be related to the awareness that wearing a mask effectively prevents the spread of COVID-19, and that its importance has been emphasized worldwide [41,42]. In fact, even before airlines provided PPE to cabin crew members, they were asked to wear PPE (e.g., goggles, aprons, gloves) during flight duty, due to concerns regarding in-flight infection. A previous study reported that wearing a mask could reduce the risk of respiratory virus transmission by 65% overall, and by 96% specifically for COVID-19 (SARS 74%, influenza 55%) [43].

Notably, we found that the average HH performance score was 4.47, which was lower than that reported for medical staff (4.67) during the COVID-19 pandemic [44]. This could potentially be related to a misconception among crew members that contaminated hands are a lesser source of respiratory infection than droplets [45]. Cabin crew, who are non-medical personnel, may have perceived HH as less important for preventing respiratory infections. However, this is unlikely, as HH awareness was significantly higher than HH performance. This discrepancy may have been due to the cabin environment being inadequate for performing frequent HH. Since the cabin crew members share a limited number of bathrooms with passengers, it could be difficult for them to regularly wash their hands with soap and water during flights. This was supported by a previous study, which found that poor sink accessibility is related to decreased hand washing [46]. To compensate for this, the airline equipped all galleys with hand sanitizer and emphasized HH; however, it still may not have been easy to perform HH during flight duty. Moreover, the cabin crew might be reluctant to use alcohol-based hand sanitizer because it can dry out the skin [47], and the incidence of dermatitis may increase due to frequent use of hand sanitizer in the cabin’s low-humidity environment, compared to on the ground [47,48]. Since HH is crucial to preventing infection transmission [49], it is necessary for airlines to provide hand moisturizer as well as hand sanitizer, and emphasize its use, to avoid dermatitis and improve HH performance. In addition, to prevent infection, this airline has introduced various procedures, such as entry screening, temperature screening, wearing masks, and social distancing, which were implemented by most airlines. Moreover, cabin crew regularly disinfect sanitary facilities during flight [30].

We found that simulation-based PPE training experience and IP awareness were related to IP performance among cabin crew. Further, although organizational culture was correlated with IP performance (r = 0.26, *p* < 0.05), this was not found in the multivariate analysis. This could be because IP awareness (r = 0.50, *p* < 0.05) was more related to IP performance than organizational culture. Therefore, IP performance could be more effectively increased through IP awareness than through organizational culture.

In this study, simulation-based PPE training experience was associated with IP performance among cabin crew. This was consistent with previous studies in which practical training was found to improve IP performance among medical staff during the COVID-19 pandemic [23,50]. This airline operated chartered flights to Wuhan, China, in January 2020, and to Milan, Italy, in March 2020. The cabin crew onboard these chartered flights received training directly from epidemiological intelligence service officers on how to put on and remove PPE, such as Level D protective suits, N95 masks, gloves, and goggles. It has been reported that simulation-based education, including specific training in IP performance, is more effective than lecture-based education focused on knowledge transfer [24]. In fact, we found that cabin crew members with simulation-based PPE training experience demonstrated HH performance after removing a Level D protective suit that was significantly superior to what is generally seen in non-medical personnel, although average HH performance was still low for all participants. Self-contamination of one’s hands after removing PPE inappropriately has reportedly occurred among medical staff [51,52]. It is, therefore, important for cabin crew members to perform HH to reduce infection risk after removing PPE. Thus, regular training is necessary to prepare for any outbreaks, and PPE training for cabin crew could be helpful in developing and employing simulation-based strategies.

In this study, IP awareness showed a strong positive correlation with IP performance (r = 0.50, *p* < 0.05), and remained a significant factor affecting IP performance in the multivariate analysis. These results were consistent with those of previous studies conducted among medical staff [19,49]. Among medical staff, having IP awareness was reported to contribute to reduced infection rates in hospitals by facilitating reflection on IP practices, willingness to engage in IP improvement programs, and increasing knowledge levels [53]. IP awareness is considered to be important for cabin crew to retain their ability to introduce and improve safety behaviors to prevent unsafe situations during flight [54]. IP awareness could be improved through IP training [19]; however, cabin crew members had few opportunities to develop skills and attend structural educational courses on preparedness for endemic or epidemic infectious diseases, such as malaria, yellow fever, Ebola, and COVID-19 [55]. According to the Aviation Act, cabin crew should receive initial and periodic qualification training for flight duty [56,57]. These programs need to include specific IP performance training to improve awareness among cabin crew regarding infectious diseases. Further, it is necessary to conduct regular trainings to raise IP awareness among cabin crew to improve IP performance, thus leading to reduced in-flight infection transmission.

The results suggest that airlines must realize the importance of cabin crew’s IP behavior to prevent global transmission of emerging infectious diseases such as COVID-19, and to offer administrative support in the form of providing hand sanitizer and PPE, as well as cabin crew’s training regarding PPE, after establishment of the airline’s infection control policy.

This study provides a new perspective on IP performance among cabin crew members; however, it also has a few limitations. First, there can be limitations in generalizing the results, as all participants were from one airline. Second, there is a possibility of overestimation due to the use of self-report measures. Third, experience of handling passengers with COVID-19 or MERS could not have reached significance, as few participants reported having such experience.

## 5. Conclusions

In this study, cabin crew demonstrated a high level of IP performance. In particular, mask-wearing was high compared to HH. Additionally, the study identifies that simulation-based PPE training experience and IP awareness are related to IP performance. To prepare for potential future pandemics which could be globally transmitted by air travel, it is necessary for airlines to develop a positive organizational culture, provide regular IP training for cabin crew members, and offer administrative support, such as proper equipment (e.g., PPE, hand sanitizer, and hand moisturizer). Additionally, it is the responsibility of cabin crew members to make an effort to improve IP performance.

## Figures and Tables

**Table 1 ijerph-18-06468-t001:** General characteristics of participants.

*(N = 177)*		
Variables	Categories	M ± SD or *n* (%)
Gender	Female	133 (75.1)
	Male	44 (24.9)
Age (years)		37.58 ± 8.46
	20~29	45 (25.4)
	30~39	57 (32.2)
	40~49	69 (39.0)
	50~59	6 (3.4)
Marriage	Married	67 (49.2)
Living with	Alone	35 (19.8)
Education	College	27 (15.3)
	University	131 (74.0)
	Graduate or above	19 (10.7)
Position	Team leader	72 (50.7)
	Staff	105 (59.3)
Working period (years)		6.26 ± 3.38
Experience handling passengers with	MERS ^1^	14 (7.9)
	COVID-19 ^2^	25 (14.1)
Quarantined due to contact with	MERS ^1^	3 (1.7)
	COVID-19 ^2^	19 (10.7)
Simulation-based PPE ^3^ training		23 (13.0)

^1^ MERS = middle-east respiratory syndrome; ^2^ COVID-19 = coronavirus disease 2019; ^3^ PPE = personal protective equipment.

**Table 2 ijerph-18-06468-t002:** Awareness and performance of infection prevention in participants.

*(N = 177)*				
Categories	M ± SD		*t*	*p*
	Awareness	Performance		
Hand hygiene	4.61 ± 0.45	4.47 ± 0.56	3.86	<0.05
Wearing a mask	4.78 ± 0.35	4.73 ± 0.35	1.72	0.088
Handling passengers with confirmed or suspected COVID-19 ^1^	4.97 ± 0.08	4.90 ± 0.41	0.96	0.347
Average	4.75 ± 0.28	4.56 ± 0.44	6.54	<0.05

^1^ COVID-19 = coronavirus disease 2019.

**Table 3 ijerph-18-06468-t003:** Univariate analysis of factors affecting infection prevention performance.

*(N = 177)*					
Variables	Categories		Performance ^5^		
			M ± SD or Median (Range)	*t, F, Z,* or r	*p*
Gender	Female		4.55 ± 0.47	−0.86	0.393
	Male		4.61 ± 0.33		
Age (years)	20~29		4.55 ± 0.44	0.60	0.616
	30~39		4.52 ± 0.42		
	40~49		4.58 ± 0.45		
	50~59		4.76 ± 0.33		
Marriage	Single		4.58 ± 0.42	0.54	0.592
	Married		4.55 ± 0.45		
Education	College		4.56 ± 0.49	0.04	0.965
	University		4.57 ± 0.43		
	Graduate or above		4.54 ± 0.38		
Position	Team leader		4.57 ± 0.43	−0.09	0.928
	Staff		4.56 ± 0.44		
Experience handling passengers with	MERS ^1^	Yes	4.87 (4.19, 5.00)	−1.76	0.078
		No	4.67 (2.79, 5.00)		
	COVID-19 ^2^	Yes	4.76 (3.33, 5.00)	−1.21	0.227
		No	4.65 (2.79, 5.00)		
Quarantined due to contact with	MERS ^1^	Yes	4.73 (3.53, 5.00)	−0.03	0.979
		No	4.67 (2.79, 5.00)		
	COVID-19 ^2^	Yes	4.76 (3.33, 5.00)	−0.92	0.361
		No	4.67 (2.79, 5.00)		
Simulation-based PPE ^3^ training	Yes		4.76 (3.33, 5.00)	−1.99	< 0.05
	No		4.64 (2.79, 5.00)		
IP awareness (range 1–5) ^4^			4.75 ± 0.28	0.50	< 0.05
Organizational culture (range 1–7) ^4^			5.96 ± 0.77	0.26	< 0.05

^1^ MERS = middle-east respiratory syndrome; ^2^ COVID-19 = coronavirus disease 2019; ^3^ PPE = personal protective equipment; ^4^ Evaluated using Pearson’s correlation coefficient; ^5^ Calculated using non-parametric analysis of Mann-Whitney U test due to small responses (under 30 cases) in variables of experience handling possible infected passengers or quarantined due to contact with suspected infectious passengers, or PPE training.

**Table 4 ijerph-18-06468-t004:** Multivariate analysis of factors affecting infection prevention performance using linear regression with bootstrapping method.

(*N* = 177)						
Variables ^1^	B	Bootstrap			Clearance	VIF ^4^
		SE	95% CI ^3^	*p*		
Constant	0.40	0.54	−0.75–1.41	0.475		
Awareness	0.77	0.11	0.54–1.03	<0.05	0.84	1.19
Simulation-based PPE training ^2^	0.25	0.07	0.11–0.39	<0.05	0.97	1.03
Organizational culture	0.04	0.05	−0.40–0.13	0.347	0.79	1.27
*Adj R*^2^= 0.29, F = 12.84 (*p* < 0.05), Durbin-Watson’s d = 1.97						
Kolmogorov-Smirnov normality test Z = 1.95 (*p* = <0.05), Breusch Pegan test χ^2^ = 16.80 (*p* = <0.05)						

^1^ Adjusting age, working period, and position; ^2^ Dummy variable = Yes or No (reference); ^3^ Confidence interval is calculated after correcting for bias; ^4^ VIF = variance inflation factor.

## Data Availability

The data are not publicly available due to privacy or ethical restrictions.

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
