# Peer review of "Infection Prevention Performance among In-Flight Cabin Crew in South Korea"

_ijerph, 2021, doi:10.3390/ijerph18126468_

Round 1

Reviewer 1 Report

The article presents data regarding an online survey with 177 cabin crew  members between August and September 2020. The survey assessed IP performance, IP awareness,  simulation-based personal protective equipment (PPE) training experience, and organizational culture. The article presents interesting information however I did not identify a solid conclusion. Did the participants tested for COVID? are any of the results associated with the interview? How do you change the person culture? What was the most identified problem involved with the passengers not taking care of the COVID pandemics? 

Reviewer 2 Report

Comments to the Authors

An interesting and prospecting paper, which is worth of accepting after the recommended changes.

Remarks and questuons

Abstract

In this section the authors write about performance-score without explaining what it exactly means. It would be very necessary to make it clear, because without this information, the whole Abstract sections is not able to fullfil its mission.

Intro

In this section I miss the summary of  general infection preventions steps which have been being used in the last decades. It would be also useful to write briefly about such protocols of other flight companies. These informations would help a lot to see the significance of the actual changes in infection prevention, due COVID-19 pandemy.

The authors mention the MERS epidemy in 2015; what changes were induced in the infection prevention after that epidemy?

Materials and Methods

Unfortunately the authors do not discuss the size of the cabin, the volume of the air content, compared to the number of passengers and crew members, though it would be important, as the the different aircraft types may make different steps of infection prevention necessary. If not, than it must be communicated.

Results

Good, detailed, infromative. In Table 1, the authors show in formations about female participants, why?

Discussion

In this section a certain kind of comparison to the infection prevention protocol of other flight companies would be interesting. If such in formations are available, they would be worth of mentioning briefly.

References

The number of the references is too low. I miss references from different countries of the region dealing with the same issue.

For example:

G. Kemenesi et al. Multiple SARS-CoV-2 introductions shaped the early outbreak in Cebntral Eastern Europe: comparing Hungarian data to a worldwide sequence data-matrix. Viruses 2020; 12, 1401

Z. Szabó et al. The potential beneficial effect of EPA and DHA supplementation managing cytokine storm in Coronavirus disease. Frontiers in Physiology 11. 752. 2020.

Reviewer 3 Report

I found this investigation fascinating and important.  It is a really good study...AND i think the signals detected by the research group are likely ACTUALLY present...

BUT... I have some concerns.

Specifically the use of parametric analysis with data tht are very arguably non-parametric.  The Chronbachs alpha is low...too low...

I highly recommend a different analytical approach.  Using 'means' of nonparametric variables is not a great idea.  I would encourage non-parametric analysis instead.  I think it is inapprpriate to use Likert questions as parametric variables (e.g. using means to tests associations...t tests). Likert scales behave like parametric variables, likert questions do not.  Plus, the scale (with a Chronbach's of 0.64) is VERY tenuous indeed.

This article (Carifio, J., & Perla, R. (2008). Resolving the 50‐year debate around using and misusing Likert scales. Medical education42(12), 1150-1152.) is a great article that makes a great case for using parametric analyses on TRUE Likert scales...note the last page where is states parametric analysis should be used only exceeding rarely on 'Likert Items'.  I worry that your scale is not truly a scale and you should be treating the items as non-parametric data.

One more small critique (and it may just be a pet peeve of mine):  Report p as being less than your critical value (i.e. p<0.05).  the size of p doesnt really matter once we reject our null hypothesis.  Using p as a replacement for effect size is something we need to move past as a scientific community.

NOW...overall, I loved the article and found it fascinating. I hope these comments are helpful.  I really liked the discussion.

Round 2

Reviewer 3 Report

This is much improved.  It is a great article and will add to our understanding.  I learned a great deal from reviewing it.  Glad the Cronbach's alpha improved with the removal of that superfluous item.